# Polyphenolics from *Syzygium brachythyrsum* Inhibits Oxidized Low-Density Lipoprotein-Induced Macrophage-Derived Foam Cell Formation and Inflammation

**DOI:** 10.3390/foods11213543

**Published:** 2022-11-07

**Authors:** Xue-Lian Chen, Pu-Lin Liang, Ming-Jiong Gong, Ya Xu, Liang Zhang, Xiao-Hui Qiu, Jing Zhang, Zhi-Hai Huang, Wen Xu

**Affiliations:** 1Key Laboratory of Quality Evaluation of Chinese Medicine of the Guangdong Provincial Medical Products Administration, The Second Clinical College, Guangzhou University of Chinese Medicine, Guangzhou 510006, China; 2State Key Laboratory of Tea Plant Biology and Utilization, Anhui Agricultural University, Hefei 230036, China; 3Department Pharmaceutical Biosciences, Uppsala University, Uppsala 75123, Sweden

**Keywords:** *Syzygium brachythyrsum*, polyphenols, foam cell, ox-LDL, cholesterol, inflammation

## Abstract

Evidence suggests that the immunomodulatory property of polyphenols may also contribute to the reduction of cardiovascular risk. In the present study, we investigated the polyphenol extraction (PE) from *Syzygium brachythyrsum*, a functional food resource in south China, regarding the protective effect on inhibiting foam cell formation and the underlying molecular mechanism based on an ox-LDL-induced RAW264.7 macrophage model. The results of Oil Red O staining, Dil-ox-LDL fluorescent staining, and cholesterol efflux experiments showed that PE, and its two phenolics brachythol B (BB) and ethyl gallate (EG), significantly inhibited the foam cell formation, which may be associated with reducing the expression of SR-A1 and CD36 while increasing expression of SR-B1, ABCG1, and PPARγ. In addition, BB and EG also reduce the inflammatory response by down-regulating the expression of NF-κB and MAPK signal pathway proteins, thereby inhibiting the expression of inflammatory factors. Altogether, PE and its two components BB and EG attenuated foam cell formation and macrophage inflammation response.

## 1. Introduction

Atherosclerosis (AS) is a chronic, complex inflammatory cardiovascular disease that can lead to critical problems, including heart attack, stroke, or even death. During the early stage of AS progress, the formation of foam cells plays a vital role [1]. The imbalance of oxidized LDL (ox-LDL) influx, esterification, and efflux causes excessive accumulation of cholesterol, triggering the conversion of macrophages to foam cells [2]. Scavenger receptor class A type 1 (SR-A1) and the cluster of differentiation 36 (CD36) on macrophage cell membranes are major receptors to bind and uptake the ox-LDL through selective phagocytosis and pinocytosis, thereby mediating cholesterol influx. The excess esterified cholesterol in cells is exported mainly via reverse cholesterol transport (RCT) systems such as SR-B1 and ATP-binding cassette transporter G1 (ABCG1) [3]. Peroxisome proliferator-activated receptor-gamma (PPARγ), as a classical signaling pathway, plays a critical role in lipid removal by inducing ABCG1/SR-B1 expression to promote high-density lipoprotein (HDL)-mediated cholesterol efflux from macrophages [4]. Targeting these surface receptors is believed to be an effective strategy for regulating lipid metabolism in macrophages and preventing AS.

Inflammation is another vital contributor to AS [5]. The inflammatory cytokines, including interleukin (IL)-1β, IL-6, and tumor necrosis factor (TNF-α), activate endothelial cells, which further recruit inflammatory cells into the endothelium [6]. Meanwhile, driven by chemokines, pro-inflammatory monocytes migrate cross the artery wall, and differentiate into macrophages, exacerbating foam cell formation and intravascular aggregation [7]. Therefore, reducing macrophage inflammation is also essential in the treatment of AS.

Epidemiological studies have demonstrated that dietary consumption of polyphenol-rich natural products can reduce cardiovascular events both in the general population and in patients at risk of cardiovascular disease [8,9,10]. In vivo experiments suggested that some polyphenols can protect against AS-related diseases by regulating lipid uptake and efflux receptors [11].

The fruits and leaves of *Syzygium brachythyrsum*, known as wild holly fruit in China, have a long history of consumption, while the leaves are often used as functional tea and folk medicine in south and southwest China [12,13]. According to the traditional Chinese medicine theory, it relieves asthma and moistens the lungs, etc. [14]. However, few studies have been reported on this species, and its potential bioactive characteristics remain under-explored. Our previous chemical investigation has demonstrated that it contains a large amount of gallic acid-type polyphenols, of which most are bergenin derivatives, and its polyphenol extraction (PE) and the isolated polyphenols were also shown to possess decent antioxidant and anti-inflammatory activities [15]. Polyphenols such as gallic acid derivatives and bergenin derivatives are generally characterized by the presence of hydroxyl groups attached to one or more aromatic rings and show increasing evidence of beneficial effects [16,17,18]. However, the precise role of these polyphenolics in the prevention of AS-related diseases remains unknown.

As an ongoing program, we are dedicated to the screening of natural dietary polyphenolic products for preventing cardiovascular risk at an early stage and our preliminary screening test indicated that the polyphenol extraction (PE), as well as some isolated phenolic compounds, could reduce the lipid uptake in the RAW 264.7 cell model. In the present study, we highlight their potency to inhibit the formation of the macrophage foam cell and the underlying mechanisms. The results will help provide a theoretical basis for the development of botanical nutraceuticals beneficial to cardiovascular health.

## 2. Methods

### 2.1. Materials

The leaves of *S. brachythyrsum* were collected in Xishuangbanna Dai Nationality Autonomous Prefecture, Yunnan Province, China.

### 2.2. Extractions and Isolations

The preparation of PE and the isolation of thirteen polyphenolics (Figure 1) were conducted in our previous study [15]. These compounds, including seven bergenin derivatives (**1**–**6**, **8**), two flavonoids (**9**–**10**), and four gallic acid derivatives (**7**, **11**–**13**) isolated from the PE fraction of *S. brachythyrsum* by column chromatography (CC) and semi-preparative HPLC, were structurally proved by NMR and LC-MS spectroscopic methods. Compound **1** brachythol A (BA) and **2** brachythol B (BB) are two new chemicals and all other polyphenolics, including 11-O-galloylbergenin, 11-O-caffeoylbergenin, 11-O-(E)-ferulate, bergenin, ellagic acid, valoneaic aciddilactone, kaempferol, quercetin, gallic acid, ethyl gallate (EG), and gentisic acid, were isolated from the Syzygium genus for the first time. Further LC-MS analysis led to the characterization of 107 polyphenolic compounds, including 9 flavonoids, 48 phenolic acids, and 50 bergenin derivatives, revealing that *S. brachythyrsum* is an excellent source of bergenin derivatives and other gallic acid-type polyphenolics. The methods of isolation and characterization of PE have been depicted in our previous work [15].

### 2.3. Cell Culture

Murine RAW264.7 macrophages (Cell Bank of Shanghai of Chinese Academy of Sciences) were cultured with DMEM containing penicillin–streptomycin solution (1%) and fetal bovine serum (10%) at 37 °C and 5% CO_2_ in a thermostat. Cells were stimulated with ox-LDL (80 μg/mL, Yiyuan Biotechnology)/LPS (1 μg/mL, Sigma Aldrich, St. Louis, MO, USA) with or without PE and its isolated compounds. Dexamethasone (DXMS, 2 μg/mL) served as a positive drug for the LPS (1 μg/mL)-stimulated inflammatory model. The treated samples were diluted into DMEM solution to various working concentrations (final concentration of dimethyl sulfoxide < 0.1% (*v*/*v*)). The doses of PE and isolated polyphenols were determined according to our previous MTT colorimetric assay [15].

In brief, the final concentrations of PE were 10, 5, and 2.5 μg/mL. The final concentrations of the isolated compound valoneaic acid dilactone were 100, 50, and 25 μmol/mL, those of quercetin, gallic acid, brachythol A, brachythol B, and 11-O-galloybergenin were 50, 25, and 12.5 μmol/mL, those of kaempferol were 30, 15, and 7.5 μmol/mL, those of ethyl gallate were 25, 12.5, and 6.25 μmol/mL, and those of ellagic acid were 5, 2.5, and 1.25 μmol/mL [15].

### 2.4. Oil Red O Staining

The formation of foam cells and the accumulation of intracellular lipid droplets were detected by Oil Red O staining. RAW264.7 macrophages were seeded into 24-well sterile culture plates and co-cultured with ox-LDL (80 μg/mL) and different concentrations of drugs for 24 h. Then, 4% paraformaldehyde was added to fix the cells for 30–40 min, and then an Oil Red O working solution (Macklin Biochemical, Shanghai, China) was applied to the attached cells for 15–20 min. Finally, images of the intracellular lipid droplets were collected with a microscope (Nikon, Tokyo, Japan) [19].

### 2.5. Dil-ox-LDL Uptake Determination

The fluorescence intensity and area of 1,1′-dioctadecyl-3,3,3′,3′-tetra-methylindocarbocyanine perchlorate (Dil)-ox-LDL were used to examine the extent of ox-LDL accumulation in RAW264.7 cells. After cell confluence, Dil-ox-LDL (20 μg/mL, Yiyuan Biotechnology Co., Ltd., Guangzhou, China) was co-cultured with different concentrations of PE and its isolated polyphenols BB and EG at 37 °C for 4 h, cell supernatant was discarded, and then washed twice with PBS. The uptake of macrophage lipids was observed under an inverted fluorescent microscope (Nikon, Tokyo, Japan). Dil-ox-DL uptake by cells was quantified using flow cytometry under the same treatment conditions to reflect the Dil-ox-DL accumulation in RAW264.7 cells. The intracellular Dil-ox-DL was calculated from the geometric mean fluorescence intensity (MFI) of 1 × 10^4^ cells (events) [20].

### 2.6. Cholesterol Efflux Assay

RAW264.7 macrophages were co-incubated with ox-LDL and 22-NBD-cholesterol (5 μg/mL, Sigma Aldrich, St. Louis, MO, USA) for 24 h. The supernatant was discarded and RAW264.7 cells were washed with PBS, then incubated with HDL (50 μg/mL) and different concentrations of PE, BB, and EG in a phenol red-free medium for 24 h. Cell supernatant was collected and cells were lysed with RIPA on ice. The fluorescence intensity of cell lysate and cell supernatant was quantified (excitation wavelength = 469 nm, emission wavelength = 538 nm). Cholesterol efflux rate = supernatant count/(cell lysate count + supernatant count) × 100% [20].

### 2.7. RT-qPCR

To perform the analysis of mRNA expression, cells (5 × 10^5^ cells/well) were treated with BB (50, 25, 12.5 μmol/mL) and EG (25, 12.5, 6.25 μmol/mL) and co-incubated with ox-LDL for 24 h. Total mRNA was extracted from cell culture with TRIzol after treatment. Equal amounts of RNA were synthesized into cDNA using the Transcriptor First Strand cDNA Synthesis 177 Kit (Roche). The quantitative PCR primers and Roche FastStart Universal SYBR Green Master Kit 179 (Roche) were adopted for real-time quantification of the gene expression of CD36, SR-A1, ABCG1, SR-B1, IL-6,IL-1β, and TNF-α. The primer genes (Table 1) were designed by Primer-BLAST (NCBI, NIH) and synthesized by Sangon Biotech (Shanghai, China). Reverse transcription reaction procedure: add 4 × gDNA Wiper Mix, running at 42 °C for 2 min; then add 5 × HiScript III RT SuperMix, running for 15 min at 37 °C; 85 °C for 5 s, and finally cool down to 4 °C for cycling. PCR reaction program: 95 °C for 10 min; 95 °C for 15 s; 60 °C for 60 s, total of 40 cycles. Calculation: normalization with GAPDH gene and determination of cycle threshold (Ct), relative mRNA expression of target genes according to the 2^−ΔΔCt^ method [21].

### 2.8. Western Blot Assay

After being treated with ox-LDL and co-incubated with or without BB (50, 25, 12.5 μmol/mL) and EG (25, 12.5, 6.25 μmol/mL) for 24 h, cellular proteins were extracted with RIPA solution for 30 min and the protein concentrations were then quantified by a BCA protein assay kit (Beyotime). Equal cellular proteins were firstly separated by 8% SDS-PAGE gels and then transferred to polyvinylidene difluoride (PVDF) membranes in a pre-cooled transfer solution. After blocking with 5% bovine serum protein for 1.5 h, membranes were incubated overnight at 4 °C with the following primary antibodies: ABCG1 and SR-B1 (Abcam, Cambridge, UK), p-P38, P38, p-P65, P65, GAPDH, β-tubulin (Cell Signaling Technology Inc., Danvers, MA, USA), PPARγ, SR-A1, CD36, P-ERK1/2 ERK1/2, P-JNK, JNK, TLR4 (Affinity Biosciences, Zhenjiang, China) at 1:1000 dilution. Then, membranes were incubated with HRP-conjugated secondary antibody (New England Biolabs, Ipswich, MA, USA) at 1:5000 dilution for 1 h. Finally, enhanced chemiluminescence (ECL) solutions were used to enhance the signal of specific proteins [22].

### 2.9. Statistical Analysis

The experimental data were expressed as mean ± SD, and analyses were performed using SPSS Statistics 26.0. Firstly, if the normal distribution requirement was met by the normality test, the one-way analysis of variance (ANOVA) parametric test was used, and the F-test was also performed. Assuming that the variance was homogeneous (*p* > 0.05), comparison between the groups was performed based on the results of LSD; if the variance was not homogeneous (*p* < 0.05), the differences between groups were evaluated based on Dunnett’s T3 results. If the normal distribution was not met, the Mann–Whitney U nonparametric test was used to analyze the variance of two independent samples. Images were plotted using GraphPad Prism 9 software. A value of (*) *p* < 0.05 was considered statistically significant, and (**) *p* < 0.01 was considered highly statistically significant. Each experiment was performed independently at least three times.

## 3. Results

### 3.1. Effects of PE and Isolated Compounds on ox-LDL-Induced Foam Cell Formation

Firstly, an Oil Red O staining assay was adopted to evaluate the effect of PE and its isolated compounds on lipid accumulation in macrophages by microscopic imaging. Lipid droplet staining excessively increased after being loaded with ox-LDL compared to the untreated control group, indicating that foam cells had been established (Figure 2). However, pre-treatment with PE effectively reduced foam cell formation in a dose-dependent manner (Figure 2A). This assay also revealed that two isolated polyphenolics, BB and EG (Comp. **1**, **12**), displayed the highest level of inhibitory effects on macrophage lipid accumulation and foam cell formation (Figure 2), while gallic acid, valoneaic acid dilactone, kaempferol, quercetin, and 11-O-(E)-ferulatebergenin (Comp. **5**, **8**–**11**) displayed inferior inhibitory activities compared to BB and EG. Other compounds failed to display lipid accumulation inhibitory activity.

### 3.2. The Effect of PE and Isolated Compounds on ox-LDL Uptake

Excessive uptake of ox-LDL by cells is a signal of foam cell formation. Dil-labeled ox-LDL can be vividly observed under fluorescence microscopy to determine the extent of ox-LDL uptake and can also be assessed quantitatively by flow cytometry. In the present study, Dil-ox-LDL significantly promoted foam cell formation, evidenced by the formation of the red fluorescence area and intensity in the positive control (Figure 3A,B), while this increase was dramatically reversed when treated with PE and two isolated compounds, BB and EG, at three concentrations. These effects were also quantitatively investigated using flow cytometry analysis (Figure 3C1–C3), where the content of intracellular Dil-ox-DL of each group was calculated by MFI. Of note, EG was shown to have better inhibitory effects on lipid accumulation.

Oil Red O stain and Dil-ox-LDL uptake assessment confirmed that BB and EG, the two isolated compounds from PE, affect the inhibition of foam cell formation by decreasing the lipid deposition in RAW264.7 cells. Therefore, BB and EG were chosen for further study.

### 3.3. The Effect of PE and Its Two Components BB and EG on Cholesterol Efflux

The accumulated lipids in macrophages can also be improved by promoting cholesterol efflux [21]. To this end, we examined whether PE and its bioactive components BB and EG have similar effects on NBD-cholesterol-loaded macrophages stimulated with ox-LDL and analyzed the HDL-mediated efflux of marked cholesterol. We found that treatment with ox-LDL significantly suppressed the HDL-mediated cholesterol outflow compared to the control group. In contrast, PE and its bioactive components BB and EG reversed ox-LDL-inhibited cholesterol efflux notably (Figure 4), where EG at 25 μmol/mL (80.21%) showed an excellent cholesterol efflux effect in macrophages. These results indicated that promoting cholesterol efflux may be another pathway for PE, BB, and EG to inhibit foam cell formation.

### 3.4. BB and EG Regulated Cholesterol Homeostasis

Supplementation with BB and EG caused a remarkable decrease in the expression of SR-A1 and CD36 at the transcriptional level (Figure 5), implying that BB and EG might be able to inhibit ox-LDL-induced foam cell formation by reducing lipoprotein internalization. This assumption was further supported by Western blot assay. As shown in Figure 5, ox-LDL-mediated macrophages could trigger higher expression of CD36 and SR-A1 proteins. Conversely, BB significantly suppressed this condition at 25 and 50 μmol/mL (Figure 5A-2,B-2), whereas EG markedly suppressed SR-A1 expression at 12.5 and 25 μmol/mL (Figure 5C-2) and inhibited CD36 (D-2) expression in a dose-dependent manner.

Excessive free cholesterol (FC) outflow to extracellular areas can be promoted by HDL-mediated ABCG1 and SR-B1 pathways. This experiment showed a significant decrease in ABCG1 and SR-B1 mRNA expression by ox-LDL-stimulated RAW264.7 macrophages compared to the normal control group. Consistent with the results of HDL-mediated cholesterol efflux, BB (50 and 25 μmol/mL) and EG (25 and 12.5 μmol/mL) treatments significantly increased the gene expression of lipid efflux receptors ABCG1 and SR-B1. We further analyzed the receptors’ protein expression by Western blot experiments. As shown in Figure 5, the protein expression of ABCG1 and SR-B1 was notably up-regulated by BB (50 and 25 μmol/mL) and EG (25 and 12.5 μmol/mL). Additionally, PPARγ protein expression was dramatically increased when administered with BB and EG in a dose-dependent manner (Figure 5I,J).

The above result demonstrated that BB and EG treatment might maintain cholesterol homeostasis in RAW264.7 macrophages through CD36, SR-A1, and PPARγ-ABCG1/SR-B1 pathways.

### 3.5. BB and EG Regulated P65 NF-κB, TLR4 and JNK, ERK1/2, P38 MAPK Activity in ox-LDL-Stimulated RAW264.7 Cells

Evidence suggested that NF-κB and MAPK pathways are typical pro-inflammatory signaling pathways involved in inflammation and lipid accumulation [23,24]. In addition, TLR4 activation can bind to the TLR structural domain of MyD88 and trigger a series of phosphorylation reactions to activate NF-κB kinase, thereby activating NF-κB [25,26,27] and ultimately promoting the production of TNF-α, IL-6, and IL-β [28,29,30]. Based on these studies, we investigated the expression of key proteins in the NF-κB and MAPK pathways in ox-LDL-induced foam cells. As shown in Figure 6, BB and EG treatment substantially suppressed the phosphorylation of P65 NF-κB, while BB at 50 μmol/mL and EG at 25 μmol/mL treatment also prominently mitigated the protein level of TLR4. Both BB and EG resulted in the enormous reduction of JNK phosphorylation. Furthermore, the treatment with BB (50, 25 μmol/mL) and EG (25, 12.5 μmol/mL) also prominently mitigated P38 and ERK1/2 phosphorylation.

In conclusion, the data suggest that BB and EG exert a significant blocking effect on the NF-κB/MAPK pathway to attenuate foam cell formation and inflammatory responses.

### 3.6. BB and EG Inhibited LPS-Induced Inflammatory Responses

Early inflammation is involved in promoting foam cell formation. LPS leads to a significant increase in the production and mRNA expression of inflammatory cytokines such as TNF-α, IL-1β, and IL-6 in RAW264.7 macrophages. In contrast, these effects were potentially attenuated in treatment with BB and EG (Figure 7). BB could decrease the production and mRNA expression of IL-1β and IL-6 at various concentrations and potently inhibited the expression of TNF-α at 50, 25 μmol/mL. Likewise, pre-treatment of cells with EG resulted in a dose-dependent inhibitory effect on TNF-α production and mRNA expression, with roughly the maximum inhibition of IL-6 at 25 μmol/mL (93.04%), which was close to DXM (94.00%). In addition, EG at 25, 12.5 μmol/mL also showed a significant inhibition effect of IL-1β.

## 4. Discussion

Dietary supplementation of natural products with polyphenolics has been proven to be very beneficial in regulating cholesterol homeostasis in macrophages and delaying the development of AS [10,31]. Our study first demonstrated that PE from *S. brachythyrsum* and its polyphenolic compounds BB and EG alleviated the ox-LDL-induced cholesterol and lipid accumulation and inflammation in RAW264.7 macrophages. Hence, these results open a fresh avenue to investigate the potential protective effects of PE on AS and indicate that *S. brachythyrsum* may become an intriguing target for dietary intervention.

The inhibitory effect of the two phenolic constituents BB and EG on ox-LDL internalization in RAW264.7 cells might be related to their down-regulation of the expression of SR-A1 and CD36 at transcriptional and translational levels. Meanwhile, BB and EG effectively promoted HDL-mediated cholesterol efflux by remarkably up-regulating the expression of SRB1 and ABCG1. Additionally, both BB and EG significantly improved PPARγ protein expression which implied that they might elevate ABCG1 and SR-B1 protein expression via activating the PPARγ pathway [32,33].

NF-κB and MAPK are also closely associated with foam cell formation [33,34]. In previous research, p65 NF-κB and p38 MAPK are proven as critical key regulators of macrophage foaminess [35,36,37], and the c-Jun N-terminal kinase (JNK), ERK, and p38 kinase pathways are the typical pathways of MAPK signaling [38,39]. The activation of p38 MAPK was revealed to inhibit the cholesterol efflux and the expression of both SR-B1 and ABCG1 in ox-LDL-mediated macrophages [40], while p38 MAPK and JNK inhibitor remarkably suppressed foam cell formation [41]. Our current study showed a substantial decrease in P65 NF-κB, P38/JNK/ERK1/2 phosphorylation in ox-LDL-induced macrophages after BB and EG treatment, which may be associated with the relief of inflammation and the reduction in cholesterol accumulation. Meanwhile, macrophages can regulate the phagocytosis of ox-LDL via multiple response mechanisms during foam cell formation. Many natural products showed immunomodulatory activity toward immune-responder cells such as modulating the phagocytic activity of macrophages [42]. Phenolic extracts of rice bran were found to attenuate the inflammatory response by reducing malondialdehyde, nitric oxide, and pro-inflammatory cytokines [43]. The mechanism of *S. brachythyrsum* polyphenol- reduced foam cell formation via regulating macrophage functionality needs to be further investigated in the future.

In summary, polyphenolics from *S. brachythyrsum* showed inhibitory effects on macrophage foam cell formation and inflammation. The mechanism might be partly modulated by PPARγ, NF-κB, and MAPK signaling pathways. Our study suggests that it could be explored as a beneficial bioactive botanical supplement for cardiovascular health, and its mechanism in this process could be further confirmed in vivo.

## Figures and Tables

**Figure 1 foods-11-03543-f001:**
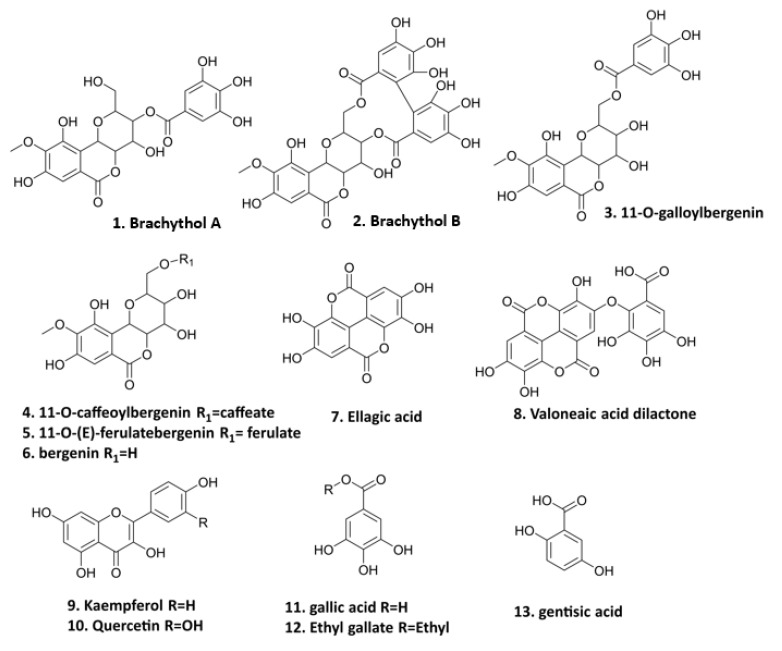
Polyphenolic compounds from PE (**1**. brachythol A; **2**. brachythol B; **3**. 11-O-galloylbergenin; **4**. 11-O-caffeoylbergenin; **5**. 11-O-(E)-ferulate; **6**. bergenin; **7**. ellagic acid; **8**. valoneaic acid dilactone; **9**. kaempferol; **10**. quercetin; **11**. gallic acid; **12**. ethyl gallate; **13**. gentisic acid).

**Figure 2 foods-11-03543-f002:**
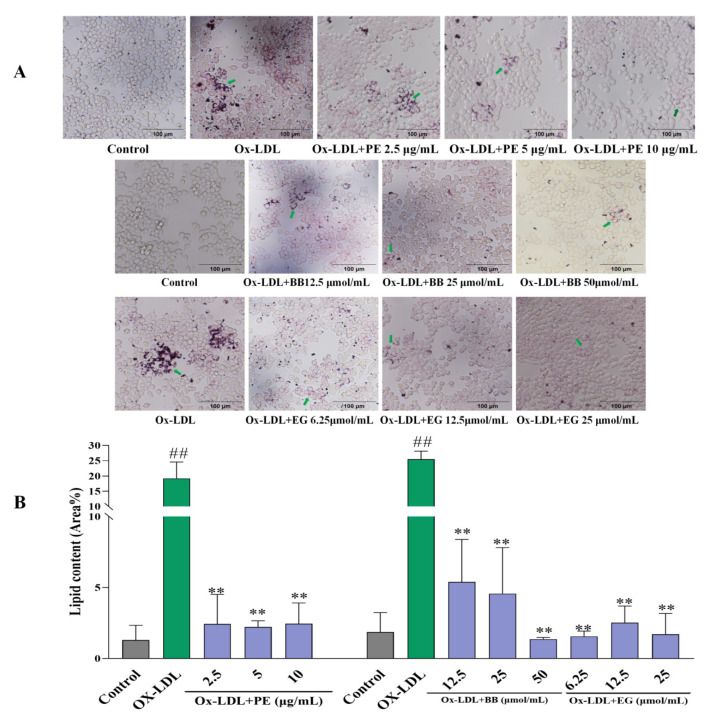
The effect of PE, BB, and EG (**A**) on lipid accumulation by Oil Red O staining and the green arrows indicates the oil red stained cellular lipids; the lipid content (Area%) was calculated by ImageJ software (**B**). All data are run in triplicate, ## *p* < 0.01 vs. the control group, ** *p* < 0.01 vs. ox-LDL-induced group.

**Figure 3 foods-11-03543-f003:**
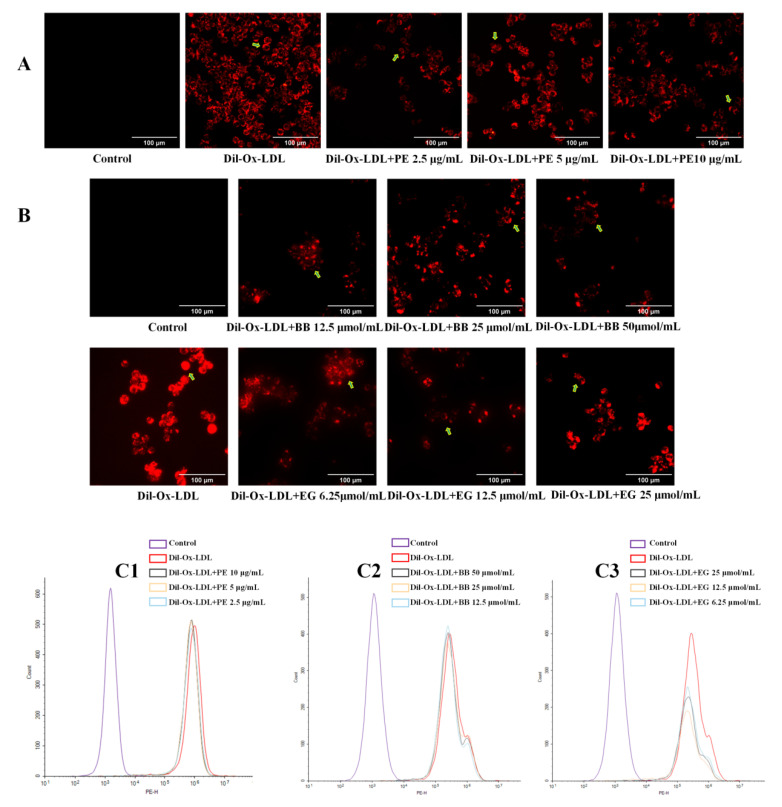
The effect of PE (10, 5, 2.5 μg/mL) and its two components BB (50, 25, 12.5 μmol/mL) and EG (25, 12.5, 6.25 μmol/mL) on Dil-ox-LDL absorption. The fluorescence staining area and intensity of the cells were observed by inverted fluorescence microscopy (**A**,**B**, the Dil-ox-LDL loaded cells were pointed out by green arrows); the contents of intracellular Dil-ox-DL were calculated by MFI with flow cytometry analysis (**C1**–**C3**). All data were run at least three times.

**Figure 4 foods-11-03543-f004:**
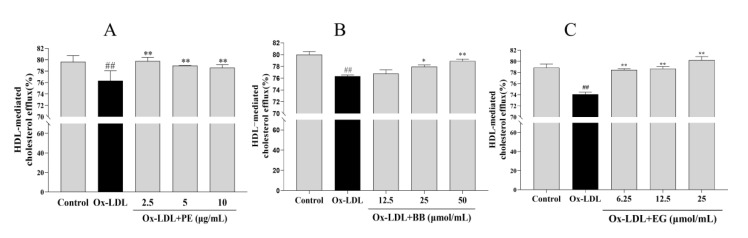
The effect of PE (**A**), BB (**B**), and EG (**C**) on cholesterol efflux mediated by HDL. ## *p* < 0.01 vs. normal control; * *p* < 0.05, ** *p* < 0.01 vs. ox-DL-induced group.

**Figure 5 foods-11-03543-f005:**
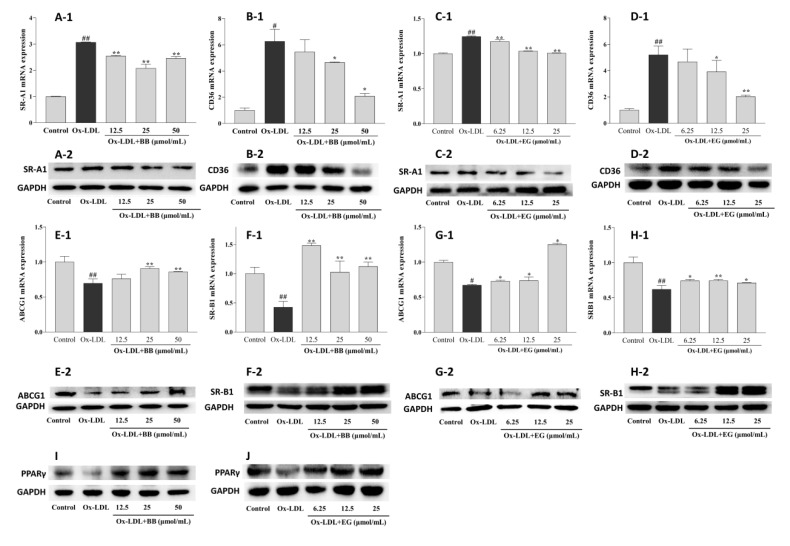
Effects of BB (12.5, 25, 50 μmol/mL) and EG (6.25, 12.5, 25, 50 μmol/mL) on SR-A1 and CD36 gene (**A-1**,**B-1**,**C-1**,**D-1**) and protein (**A-2**,**B-2**,**C-2**,**D-2**) expressions in the ox-LDL-induced RAW264.7 cells; effects of BB (12.5, 25, 50 μ mol/mL) and EG (6.25, 12.5, 25, 50 μ mol/mL) on ABCG1 and SR-B1 gene (**E-1**,**F-1**,**G-1**,**H-1**) and protein (**E-2**,**F-2**,**G-2**,**H-2**) expressions in ox-LDL-mediated RAW264.7 cells; effects of BB (12.5, 25, 50 μ mol/mL) and EG (6.25, 12.5, 25, 50 μmol/mL) on PPARγ (**I**,**J**) protein expression in RAW264.7 cells under ox-LDL (80 μg/mL) condition. # *p* < 0.05, ## *p* < 0.01 vs. the control group, and * *p* < 0.05, ** *p* < 0.01 vs. ox-LDL-induced group.

**Figure 6 foods-11-03543-f006:**
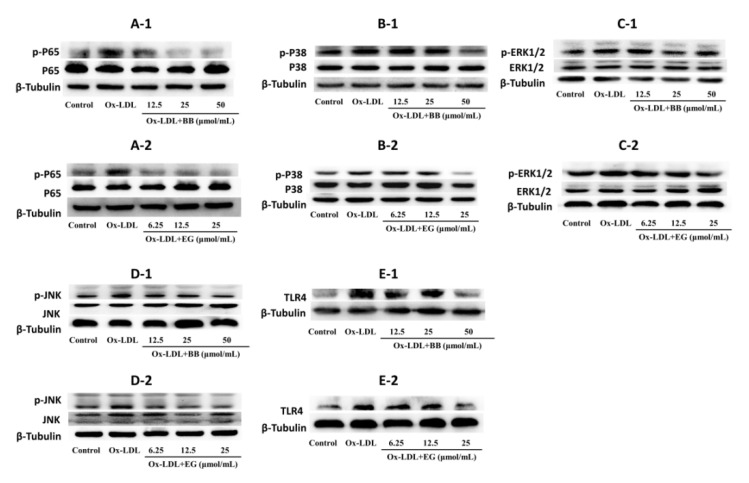
p-P65/P65 (**A-1**,**A-2**), p-P38/P38 (**B-1**,**B-2**), p-ERK1/2/ERK1/2 (**C-1**,**C-2**), p-JNK/JNK (**D-1**,**D-2**), and TLR4 (**E-1**,**E-2**) protein expression in RAW264.7 macrophages. Data were collected at least three times.

**Figure 7 foods-11-03543-f007:**
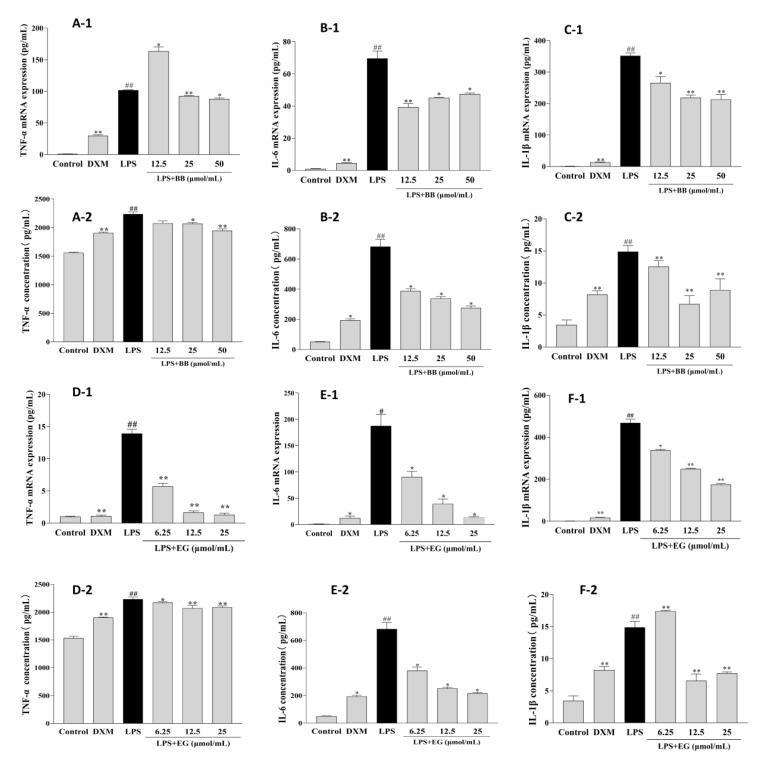
Effects of BB and EG on LPS-induced TNF-α (**A-1**,**D-1**), IL-6 (**B-1**,**E-1**), and IL-1β (**C-1**,**F-1**) mRNA expression; effects of BB and EG on LPS-stimulated accumulation of TNF-α (**A-2**,**D-2**), IL-6 (**B-2**,**E-2**), and IL-1β (**C-2**,**F-2**) in macrophages. RAW264.7 cells were incubated with LPS and co-cultured with different concentrations of BB and EG for 24 h. For ELISA measurements, the levels of TNF-α, IL-1β, and IL-6 in the cell supernatant were measured with ELISA kits. # *p* < 0.05, ## *p* < 0.01 indicates compared to control; * *p* < 0.05, ** *p* < 0.01 indicates compared to LPS-induced group.

**Table 1 foods-11-03543-t001:** The primers sequences of target genes.

Genes	Forward Primer Sequences (5′-3′)	Reverse Primer Sequences (5′-3′)
GAPDH	GGTTGTCTCCTGCGACTTCA	TGGTCCAGGGTTTCTTACTCC
SR-A1	GACACTGATAGCTGCTCCGAATCTG	AAACACGAGGAGGTAAAGGGCAATC
CD36	GTCTATCTACGCTGTGTTCGGATCTG	TGTCTGGATTCTGGAGGGGTGATG
ABCG1	CTGCTGCCTCACCTCACTGTTC	TCTCGTCTGCCTTCATCCTTCTCC
SR-B1	AGCATTCCTTGTTCCTAGACATCCATC	AACCACAGCAACGGCAGAACTAC
TNF-α	CGCTCTTCTGTCTACTGAACTTCGG	GTGGTTTGTGAGTGTGAGGGTCTG
IL-6	CTTCTTGGGACTGATGCTGGTGAC	AGTGGTATCCTCTGTGAAGTCTCCTC
IL-1β	CACTACAGGCTCCGAGATGAACAAC	TGTCGTTGCTTGGTTCTCCTTGTAC

## Data Availability

All experimental data are shown in the article.

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
