# Peer review of "Polyphenolics from Syzygium brachythyrsum Inhibits Oxidized Low-Density Lipoprotein-Induced Macrophage-Derived Foam Cell Formation and Inflammation"

_foods, 2022, doi:10.3390/foods11213543_

Round 1
Reviewer 1 Report
The manuscript 'Polyphenolics from Syzygium brachythyrsum inhibits oxidized 2 low-density lipoprotein-induced macrophage-derived foam cell 3 formation and inflammation' is well organised and provides some very interesting insight regarding the inhibitory activity of polyphenols against the macrophage foam cell formation and inflammation. However, there are several major points that impact upon the scientific quality of the manuscript:
- Several claims mentioned in the Introduction are not cited.
- No references provided in most of the methods described in the Materials and Methods.
- Statistical analysis is inadequate as no information of the performed tests is provided.
- More in depth discussion and comparison with the existing literature is needed.
- The language needs some refinement.
Author Response
Response to Reviewer 1 Comments:
Several claims mentioned in the Introduction are not cited.
Response: Two references (DOI: 10.1016/j.cell.2022.04.004; DOI: 10.1002/jcp.26429.) have been added in the Introduction paragraph to improve credibility and accuracy. Please see lines 35 and 51.
- No references provided in most of the methods described in the Materials and Methods.
Response: Added the references in the Materials and Methods sections 2.4-2.8.
- Statistical analysis is inadequate as no information of the performed tests is provided.
Response: Statistical test data and methods have been added to section 2.9.
- More in depth discussion and comparison with the existing literature is needed.
Response: Added additional discussion for the brachythyrsum polyphenols' effect on the regulation of macrophage functionality. Please follow the modification of the discussion section.
- The language needs some refinement.
Response: The manuscript was carefully revised again to avoid language errors.
Reviewer 2 Report
1. Some English grammar and spelling corrections must be done throughout the whole paper.
2. Abbreviations such CC (Line 87) should be defined at first mention and used consistently thereafter.
3. Figure 2b shows some spots in the images, I am wondering if those are drug aggregates, did authors check the aggregation properties before performing the cellular assays.
4. Italicize “in vivo” (52, 344) and “in vitro” throughout the manuscript including References.
5. µg/ml to µg/mL in line 137, use the same format throughout the paper
6. The quality of some figures (3C1-C3) are not good and should be replaced with images with the desired resolution
7. The references cited are in some instances old (references 4,7,8,9,10,11,12 and 28), it is of utmost importance to cite newest possible references, and not the ones that are 20-25 or 30 years old. Majority of cited publications must be published in the last 10 years meaning from 2012 to 2022. Exchange the old references with new ones.
Author Response
Response to Reviewer 2 Comments
- Some English grammar and spelling corrections must be done throughout the whole paper.
Response: As suggested, we have carefully scrutinized the manuscript and made corresponding revisions, as well as rewrote typos, grammatical errors, and long sentences.
- Abbreviations such CC (Line 87) should be defined at first mention and used consistently thereafter.
Response: We have defined the CC abbreviation with "column chromatography", please see line 94.
- Figure 2b shows some spots in the images, I am wondering if those are drug aggregates, did authors check the aggregation properties before performing the cellular assays.
Response: We have re-tested our drug solutions and they didn’t show any sign of aggregation properties while the drugs dissolved into the cell culture solution. We reckon that the spots in Figure 2 are not the aggregation of the drug but possibly red crystals of the oil red dye because the acetone and isopropyl alcohol used to prepare the oil red staining solution volatilize easily, resulting in precipitation of the staining solution. Nevertheless, it does not affect our observation and judgment of the lipid staining inside cells.
- Italicize “in vivo” (52, 344) and “in vitro” throughout the manuscript including References.
Response: The comment was followed up throughout the manuscript.
- µg/ml to µg/mL in line 137, use the same format throughout the paper
Response: The units were checked and corrected in the text.
- The quality of some figures (3C1-C3) are not good and should be replaced with images with the desired resolution.
Response: Figures (3C1-C3) were re-uploaded with a higher resolution. If required, it can be uploaded separately.
- The references cited are in some instances old (references 4,7,8,9,10,11,12 and 28), it is of utmost importance to cite newest possible references, and not the ones that are 20-25 or 30 years old. Majority of cited publications must be published in the last 10 years meaning from 2012 to 2022. Exchange the old references with new ones.
Response: As suggested, the references have been updated to the newest published within the last 10 years. (Please see [5] DOI: 10.1016/j.bcp.2017.04.014; [9] DOI: 10.1007/s00424-017-1941-y; [10] DOI: 10.3390/nu11010039; [12] DOI: 10.2174/1874467213666 200320153410; [13] DOI: 10.19969/j.fxcsxb.21110405; [14] doi:10.14088/j.cnki.issn0439- 8114.2014.03.016; [33] DOI: 10.1111/nbu.12278).
Reviewer 3 Report
The manuscript titled: “Polyphenolics from Syzygium brachythyrsum inhibits oxidized low-density lipoprotein-induced macrophage-derived foam cell formation and inflammation” is well written. The manuscript is based on a well-constructed scientific concept and carried out the studies are well. However, data needs to be refined in a presentable manner. The functional studies of macrophages are missing in the manuscript. The present manuscript would be benefited by addressing the points below.
Comments:
· In figure 2, the authors need to refine the figures, add the measurement bar, and point out the stained cells with arrow bars. Adding to that the quantitative representation of stained cells.
· Similarly, in figure 3, the authors need to refine the figures, add the measurement bar, and point out the stained cells with arrow bars. Adding to that the quantitative representation of stained cells.
· Authors showed that Polyphenolics regulate the proinflammatory phenotypes of macrophages; however, the manuscript lacks the macrophage functionality data, such as phagocytosis, and what is the effect of Polyphenolics, below paper will be helpful to this manuscript; kindly refer it. “Kumar VP, Prashanth KVH, Venkatesh YP. Structural analyses and immunomodulatory properties of fructo-oligosaccharides from onion (Allium cepa). Carbohydr Polym. 2015 Mar 6;117:115-122. doi: 10.1016/j.carbpol.2014.09.039.”
Author Response
Response to Reviewer 3 Comments
In figure 2, the authors need to refine the figures, add the measurement bar, and point out the stained cells with arrow bars. Adding to that the quantitative representation of stained cells.
Response: We have added a 100 μm measurement bar to each stained image and pointed out the stained cells with arrow bars for easy observation. The stained area of all stained cells was counted using Image J software and presented in graphical form. Please check the upgraded figure 2.
Similarly, in figure 3, the authors need to refine the figures, add the measurement bar, and point out the stained cells with arrow bars. Adding to that the quantitative representation of stained cells.
Response: Thank you for the suggestion. We have added the measurement bar to each staining image and pointed out the fluorescently stained cells with arrow bars for easy observation. the contents of intracellular Dil-ox-DL were quantitatively calculated by MFI with flow cytometry analysis. Please see figure 3.
Authors showed that Polyphenolics regulate the proinflammatory phenotypes of macrophages; however, the manuscript lacks the macrophage functionality data, such as phagocytosis, and what is the effect of Polyphenolics, below paper will be helpful to this manuscript; kindly refer it. “Kumar VP, Prashanth KVH, Venkatesh YP. Structural analyses and immunomodulatory properties of fructo-oligosaccharides from onion (Allium cepa). Carbohydr Polym. 2015 Mar 6;117:115-122. doi: 10.1016/j.carbpol.2014.09.039.”
Response: Thank you for the advice. We have added more discussion about the prospect of the brachythyrsum polyphenols on the regulation of macrophage functionality.
Round 2
Reviewer 1 Report
The authors have replied adequately to my comments and have made the respective amendemnts to the manuscript.
Reviewer 2 Report
Accept